# New Insight into the Gas Phase Reaction Dynamics in Pulsed Laser Deposition of Multi-Elemental Oxides

**DOI:** 10.3390/ma15144862

**Published:** 2022-07-12

**Authors:** Xiang Yao, Christof W. Schneider, Alexander Wokaun, Thomas Lippert

**Affiliations:** 1Laboratory for Multiscale Materials Experiments, Paul Scherrer Institute, 5232 Villigen, Switzerland; xiang.yao@live.cn; 2Research Division Energy and Environment, Paul Scherrer Institute, 5232 Villigen, Switzerland; alexander.wokaun@chem.ethz.ch; 3Laboratory of Inorganic Chemistry, Department of Chemistry and Applied Biosciences, ETH Zurich, 8093 Zurich, Switzerland

**Keywords:** pulsed laser ablation, thin film growth, plasma species dynamic, energy-resolved mass spectroscopy

## Abstract

The gas-phase reaction dynamics and kinetics in a laser induced plasma are very much dependent on the interactions of the evaporated target material and the background gas. For metal (M) and metal–oxygen (MO) species ablated in an Ar and O_2_ background, the expansion dynamics in O_2_ are similar to the expansion dynamics in Ar for M^+^ ions with an MO^+^ dissociation energy smaller than O_2_. This is different for metal ions with an MO^+^ dissociation energy larger than for O_2_. This study shows that the plume expansion in O_2_ differentiates itself from the expansion in Ar due to the formation of MO^+^ species. It also shows that at a high oxygen background pressure, the preferred kinetic energy range to form MO species as a result of chemical reactions in an expanding plasma, is up to 5 eV.

## 1. Introduction

Pulsed laser deposition (PLD) is regularly used to deposit complex oxides, due to its ability to congruently transfer the target composition to the deposited film [1,2]. The deposition is conducted using a pulsed KrF laser (λ = 248 nm corresponding to 4.99 eV/photon) with a pulse length of a few 10 ns in an O_2_ background gas. The wavelength, combined with the pulse length, results in a plasma plume consisting largely of monoatomic plasma species (excited, ion, and neutral species). Adding a background gas moderates the kinetic energy of the expanding plume species, forming additional ions and excited state species. In addition, a sufficiently large oxygen background partially compensates for the loss of oxygen in the as-deposited oxide films. As a consequence, most of the oxygen in the film originates from the background gas, at least for the case of *La_0.6_Sr_0.4_MnO_3_* [3]. Therefore, the interaction between the plume species and the background gas is of primary concern. As discussed in [4,5], the presence of an O_2_ background changes the hydrodynamics of the plume expansion due to scattering between plume species and O_2_ molecules. As a result, chemical reactions can occur with mainly di-atomic metal–oxygen (MO and MO^+^) species as reaction products [3,6,7,8], and the formation of MO and MO^+^ ions in the plume is strongly related to their dissociation energies [9]. This is shown in Figure 1, where the relative number of MO^+^ species (MO^+^/M^+^ + MO^+^) is plotted vs. the MO^+^ dissociation energy, with data taken at an oxygen deposition pressure of 1.5 × 10^−1^ mbar. Data are taken from [9] with Sc and Eu added to the figure. At this pressure, the results resemble a chemical equilibrium of the kind:(1)M^+^ + O_2_ ↔ MO^+^ + O; or(2)M^+^ + O_2_ ↔ M^+^ + O + O.

These equations are based on mass spectrometer analysis of the plasma composition [3], without taking excited state species into account. This equilibrium holds to a large extent at lower pressures (up to ~10^−1^ mbar), since the number of tri-atomic species is small compared with the number of mono-atomic and di-atomic species [3]. When the dissociation energy of MO^+^ is larger than 5.12 eV, the binding energy of O_2_, the reaction dynamics are shifted to the right side of Equation (1), and the formation of MO^+^ is favoured. If the MO^+^ dissociation energy is smaller than 5.12 eV, the equilibrium of the reaction moves to the left of the equation, or the second equation can be assumed. Depending on the species composition of the laser-induced plasma in an O_2_ background, either the formation of MO^+^ species is favoured, or the plume consists mainly of monoatomic (ionic) species [10,11,12]. As ions from the ablation process have a wide kinetic energy range [13,14,15,16], the question of interest is to determine the favourable kinetic energy window at which plasma species react/interact with the background gas to form neutral and ionic MO species.

To answer this question for ionic species, detailed ion energy analyses of M^+^ and MO^+^ species were conducted for plume expansion at different O_2_ pressures using energy-resolved mass spectroscopy. These results were compared with the corresponding ion energy distributions at different Ar pressures to show the difference in the expansion dynamics due to the reaction with O_2_ molecules. A series of different oxide targets were ablated and the ion energies analysed at different O_2_ pressures to demonstrate the more fundamental nature of this study for the ablation process.

## 2. Materials and Methods

Ion energy analyses of the plasma plume were performed in a dedicated UHV chamber as shown in Figure 2. An energy-resolved mass spectrometer (Hiden EQP-MS, Hiden Analytical, Warrington, UK) was installed on the chamber for the ion energy analysis. The Hiden EQP-MS system combines an electrostatic energy analyser and a quadrupole mass spectrometer in tandem to measure the ion energy distributions (IED) for each specific ion. Due to the high materials flux per laser pulse, a 30-µm diameter sampling orifice was used and positioned at 4 cm from the target surface normal to the plume expansion axis. In addition, an appropriate tuning of the lenses was performed to avoid chromatic aberration and to ensure a stable ion transmission. For the ablation of the targets, a KrF excimer laser (Lambda Physik LPX 300, Coherent, Göttingen, Germany, 20 ns, λ = 248 nm, repetition rate: 5 Hz) was used with a fluence of 2 J/cm^2^ and a laser spot of 1 × 1 mm^2^. In order to capture the transient plasma plume, the mass spectrometer was triggered and gated by a pulse generator connected to a photodiode. The pulse generator provided a TTL pulse to the mass spectrometer with a pulse width of 1 ms, which was much longer than the time the plume species required to reach the mass spectrometer. Thus, the mass spectrometer collected all the required species per trigger pulse. To record the full IED, the measurement was started at negative energy values. This is an instrumental artifact due to the applied electric field at the orifice to collect either positive or negative species. The shifted IED are shown as measured.

The ion energies were analysed for the ablation of La_0.33_Ca_0.67_MnO_3_ in O_2_ and Ar with O_2_ to investigate the gas phase reaction dynamics, and Ar as a reference to distinguish the difference between elastic and inelastic collisions of plasma species with the background gas molecules. The pressure in the chamber was varied from <10^−5^ mbar (vacuum expansion) to 5 × 10^−2^ mbar. For the same Ar and O_2_ pressure values mentioned, the nominal pressure value was calibrated with respect to air, and corresponded to the O_2_ gas environment. For Ar, however, the effective pressure was larger than the nominal value by approximately 40%, as determined experimentally.

To gain a broader overview about the energetic properties of plasma species, other oxide targets including 8YSZ (8 mol% Y_2_O_3_ fully stabilised ZrO_2_), BaTiO_3_, ScMnO_3_, and LuMnO_3_ were ablated at different pressures, and the plume expansion in O_2_ was analysed. Common to these oxide targets is that they contain elements with *E*_diss_ > O_2_ (La, Y, Zr, Ti, Lu, Sc) and *E*_diss_ < O_2_ (Mn, Ba, Sr, Ca) for MO^+^ species combining heavy with light elements.

## 3. Results and Discussion

### 3.1. Kinetic Energy Distributions of M^+^ and MO^+^ Species in O_2_ and Ar

Figure 3 and Appendix A compare the IEDs of monoatomic ions at different O_2_ and Ar pressures. The background gas Ar has a similar mass (40 amu) and kinetic diameter (340 pm, spherical) to O_2_ (32 amu, 346 pm elliptical) but is considered to be chemically inert with a comparable geometrical cross section [17]. Chemical reactions with Ar are known, but none of these known products have yet been detected [18,19]. The measured IEDs were the result of both elastic and inelastic collisions during the plume expansion in the respective atmospheric environment. While elastic collisions involve only kinematics, inelastic collisions involve a series of mechanisms ranging from electronic excitation to dissociation and formation of gas molecules, where chemical reactions are triggered, even for Ar [18,19]. Elastic and inelastic collisions occur in both O_2_ and Ar atmospheres, and to some extent, intra-plume collisions at an early stage of the plume expansion. To simplify the terms involved in this study, we classified the types of collisions into non-reactive collisions (where the plume species do not form chemical bonds with the background gas in measurable quantities, as in Ar) and reactive collisions (where the plume species form chemical bonds with the background gas, such as the formation of LaO^+^ in O_2_).

The IEDs of the Mn^+^ and La^+^ ions in O_2_ and Ar at different pressures are shown in Figure 3a–d, and Ca^+^ and LaO^+^ in S1. Other MO^+^ ions such as CaO^+^ or MnO^+^ were not measurable due to their very low total intensity, even for a vacuum ablation. This reflects the fact that during the initial ablation process and plume formation, the used photon energy of 4.99 eV was able to dissociate most bonds with an energy smaller than the photon energy. In the gas phase, di-atomic species with a bond energy larger than the photon energy can only be dissociated with a multi-photon absorption process, which has a much smaller probability to occur by ns pulses. Hence, more di-atomic species with *E*_diss_ > *E*_photon_ are measurable as part of the plume species composition for an ablation in vacuum. Additionally, M^+^ species with large binding energies tend to recombine easily during the expansion. After the interaction of plume species with the laser beam, gas phase reactions via scattering are the dominant chemical processes in the plasma plume and it should be possible to differentiate between non-reactive and reactive scattering processes depending on the reactivity of the background gas species. The dissociation energy of LaO^+^ (*m* = 154) with *E*_diss_ = 8.57 eV [20] is larger than that of O_2_, and therefore stable LaO molecules and LaO^+^ ions can be formed additionally by reactions in the plume at a higher O_2_ background pressure. On the other hand, MnO^+^ and CaO^+^ have a smaller dissociation energy than O_2_ (2.9 eV; 4.1 eV < 5.12 eV) [21], and their formation at a higher O_2_ background pressure is not favoured due to a dissociation via an enhanced scattering rate. Therefore, the pressure evolution of the IED for Mn^+^ and Ca^+^ in O_2_ should reflect the influence of non-reactive collisions in a way similar to an Ar background, where elastic scattering plays a dominant role. In general, a decrease in the total amount of M-species is expected with increasing pressure, and the number of species with a high kinetic energy should be significantly reduced. The influence of non-reactive scattering to the total number of scattering events can be deduced by comparing the corresponding IEDs in O_2_ and Ar. For La^+^, one would expect to more clearly see the influence of reactive scattering, since the formation of LaO^+^ in O_2_ would be favoured with increasing O_2_ pressure and the number of La^+^-species should drop correspondingly [3,22].

The ablation of *La_0.33_Ca_0.67_MnO_3_* in vacuum Mn^+^ species (*m* = 55) had a low energy maximum with *E*_kin,max_ < 2 eV and a significant energetic tail up to 400 eV (Figure 3). With increasing O_2_ pressure, these high kinetic energies were reduced to less than 200 eV (Figure 3a), and in Ar, a substantial reduction to around 50 eV took place (Figure 3b). In addition, the intensity of the low energy peak decreased at first, in both atmospheres, before increasing to very high intensity values at 5 × 10^−2^ mbar (see Figure 3 inserts). This was also the pressure regime where the mean free path was of the order of the target–substrate distance and smaller, and hence the influence of collision became increasingly more important. Looking at Ca^+^ (*m* = 40), the high kinetic energies (300 eV in vacuum) were reduced with increasing pressure to <100 eV in O_2_ (Figure 3e) and <50 eV in Ar (Figure 3f), and *E*_kin,vac_ > 500 eV for La^+^ (*m* = 139) to less than 200 eV in O_2_ (Figure 3c) and approx. 50 eV in Ar (Figure 3d). Analogous to Mn^+^, the intensity of the low energy Ca^+^-peak decreased at first, before increasing to a very high intensity value at 5 × 10^−2^ mbar. The situation for La^+^ was different; the intensity of the low energy peak decreased with increasing O_2_ background pressure. The pressure dependence of the IED for LaO^+^ in O_2_ (Figure 3g) showed a different feature. The IED expanded from less than 20 eV in vacuum to 50 eV with increasing O_2_ pressure. The overall intensity of the IED as well as the intensity of the low energy LaO^+^-peak increased with increasing pressure while the position of the maximum remained largely unchanged with the tendency to be reduced to smaller energies. The latter was different to the low energy part of the IED for the single ionic species where a clear shift of the position for the low energy maximum was observed.

### 3.2. Dynamic Properties of Positive Ionic Species at Small Kinetic Energies

Another observation for all the measured kinetic energy distributions was the appearance of a structure at lower kinetic energies with increasing background pressure (Figure 3, insets), in particular for the ablation in Ar. Analysing these distributions more closely by fitting energy-shifted Maxwell–Boltzmann (MB) distributions, we noted that these kinetic energy distributions could be described by two or three MB distributions, indicating that the stopping of these species fell into several kinetic energy windows [23,24]. Examples are shown in the Appendix A. One window was the typical *E*_kin,max_ below 5 eV, the second was between 5 and 10 eV, and the third was between 10 and 20 eV. For the ablation in Ar, this was the general appearance; for the ablation in O_2_, there was a difference between species with *E*_diss_ larger or smaller than 5.12 eV. For M-species with *E*_diss_ < 5.12 eV, the different energy maxima with increasing pressure was a general feature, whereas for M-species with *E*_diss_ > 5.12 eV, the kinetic energy distribution was smoother but still compatible with several MB-distributions falling into the same energy windows mentioned. The origin of these energy windows is at present not understood, but the appearance of these energy maxima was certainly the outcome of an increase in the scattering rate with increasing pressure, thereby slowing down the plasma species. The scattering-induced excitation and ionisation of plasma species requires discreet energies; the multitude of different species in the plasma have a range of different energetic requirements and the net result is a structured kinetic energy distribution. The smoother energy distribution with increasing pressure in O_2_ was most likely the result of the added chemical activity in the plasma as a result of inelastic scattering leading to excitations, ionisation and dissociation, preferably for species with *E*_diss_ < 5.12 eV.

With respect to the low energy peak for Mn^+^, Ca^+^ and La^+^, all three ions showed first a drop and then an increase in intensity with increasing Ar pressure. The same behaviour was observed for Mn^+^ and Ca^+^ in O_2_. This can be seen in Figure 3. With increasing O_2_ pressures, the amount of LaO^+^ ions increased while simultaneously the amount of La^+^ decreased. In particular, starting at 10^−2^ mbar, there was a steep rise in the LaO^+^ signal with a simultaneous drop in the La^+^ signal. This indicated an increase in the reaction rate of La species (La^+^ or La*) with oxygen species (O^+^, O* or O_2_) at higher pressures. The measured intensities for LaO^+^ were mainly for *E*_kin_ < 50 eV at all pressures; above 50 eV, the intensity contribution to the total intensity became negligible. For the vacuum ablation, the LaO^+^ signal was very weak, and the measurable kinetic energy distribution reached, at most, 10 eV. Unlike La^+^, there was a very large ion intensity at low kinetic energies and an energetic tail up to 350 eV. For the ablation at 5 × 10^−2^ mbar O_2_, the LaO^+^ signal was comparable to La^+^ in vacuum at small kinetic energies and the kinetic energy range was up to 50 eV. In addition, the kinetic energy distribution showed a clear structure not seen before. The La^+^ signal, on the other hand, dropped significantly at low kinetic energies and the high energy tail reached up to 150 eV, with no distinct maximum to be measured.

### 3.3. Scattering Kinetics for the Initial Plasma Expansion

To understand this observation, the probable origin of the La^+^ and LaO^+^ kinetic energy distributions need to be discussed. The ablation of species from a target is the result of photodissociation, photoexcitation and thermal evaporation after the target material absorbs the incoming laser light. For the used wavelength and a pulse length of 20 ns, the large majority of species measured consisted of single ions, a few dimers and a very small number of trimeric species, if any. The plasma composition will also depend on the background gas and respective pressure used [3]. The initial vapor plume is formed, starts to expand, and continues to absorb the laser light, developing into a highly dense, fast moving vapour cloud with a high pressure and temperature consisting of charged, excited and neutral species. At this stage of cloud formation, the vapour will largely consist of mono-atomic species due to the dominant dissociation mechanisms, and in front of this vapour cloud, a charged layer consisting of electrons is formed, that is responsible for the acceleration of the charged species [25,26,27]. The initial shape of the plasma plume depends on its elemental composition as well as its mass, which is reasonably well described for single elemental plumes by the double layer model [25,26,27]. A good description is still missing for multi-elemental plumes, and plume dynamics—in particular, in the turbulent pressure range between 10^−2^ and 10^−1^ mbar—becomes very complex [28,29]. With a fluence of 2 J/cm^2^ used, most of the ions were expected to be single charged [30], and that dimeric species such as metal–oxygen species could form, however, scattering events and the different dissociation mechanism would result in a mixture of mono-atomic species. Only after the termination of the laser pulse would binary species become more detectable largely due to the thermal evaporation from the target and the expansion of the vapour cloud reducing the effective species density considerably within the timeframe beyond 20 ns. This scenario would account for the observation that the kinetic energy distribution for La^+^ for a vacuum ablation had an energetic tail well beyond 300 eV, and a high intensity, since these species experience a relatively long acceleration time in the electric field [31,32,33,34,35].

The kinetic energy distribution of neutral species is not well known and was not measured for these experiments. However, it can be safely assumed that they can reach considerable energies due to the almost instantaneous phase explosion when the laser beam interacts with the target material. For a vacuum ablation, fast species reach the position of the mass spectrometer or heater, respectively, within 1 µsec, as measured using time-resolved emission spectroscopy. Those optically monitored species are excited neutral species which can easily reach kinetic energies of 600 eV or more for La I at a fluence of 2 J/cm^2^ [29]. Most other plume species will arrive within 2 to 3 µsec, including all the neutrals. The LaO^+^ signal is small because the origin of these species is largely thermal and therefore neutral LaO species are formed before some of them become ionised [36]. The available acceleration time window for these ionised species is significantly smaller, leading to a kinetic energy distribution with small kinetic energies (Appendix A). With the introduction of a background gas, the situation for the scattering kinetics starts to change. A rough estimate of the number of species removed from the target with one pulse approximately corresponds to the number of atoms equivalent to a static pressure in the mbar range (10^21^–10^22^/m^3^). As a consequence of such an initial high local density, the mean free path (mfp) becomes very short (« mm), and even at the time of arrival at the location of the mass spectrometer, the plume density must be large enough to enable a sufficient number of re-excitations via scattering of optically excited species, since the typical lifetime of excited states ranges from ns to several hundred ns. This plume related mfp is still much shorter than a nominal mfp of metres for a pressure of 10^−5^ mbar and below, and the number of scattering events until species reach the mass spectrometer is at least 10 to 20. This would also imply that even at a distance 4 cm away from the target, the hydrodynamic conditions are more viscous, rather than molecular. Hence the number of additional scattering events at a relatively low deposition pressure of 5 × 10^−3^ mbar is small, but finite, leading to the creation of additional LaO species (ions, excited, neutrals) from La and O species in a reactive atmosphere. As a consequence, the number of LaO^+^ species measured in 5 × 10^−3^ mbar O_2_ increases with an extended kinetic energy range, and at the same time the La^+^ signal drops. With increasing pressure, more of the fast species will be stopped, and at the same time, react to LaO species, leading to an overall increase in the number of measured LaO^+^ species (Appendix A). At this stage a reaction kinetic of the form La^(+,^*^)^ + O_2_ ↔ LaO^(+,^*^)^ + O^(+,^*^)^ or La^(+,^*^)^ + O^(+,^*^)^ ↔ LaO^(+,^*^)^ is conceivable. The La^+^ kinetic energy distribution at 5 × 10^−2^ mbar (Figure 3c) strongly suggests that mostly La^+^ species with a small kinetic energy were involved in the conversion to LaO^+^ species. This outlined scenario would also explain why it is difficult to measure a pressure dependence for LaO^+^ in Ar. Despite a large bond energy, the readily provided LaO^+^ species from the initial ablation process were dissociated or neutralised due to inelastic scattering, in particular, at a higher pressure. Since no new LaO^+^ was formed, the signal became too small to be measured at higher pressures.

### 3.4. Gas Phase Chemistry

For the gas-phase chemistry, it is of interest to verify which kinetic energy range is suitable for the formation of di-atomic species in general, here, in particular LaO^+^. From the scattering theory a low kinetic energy range is expected to be favourable since the scattering cross section is proportional to *E*_kin_^−2^. Therefore, La^+^ (La) species with a low kinetic energy should be the most favourable combination for the formation of LaO^+^ (LaO) at a high O_2_ background pressure.

In order to work out the influence of different types of collisions of the IEDs, it is important to consider the gas density, gas molecular mass, and the van der Waals volume of the respective gas environment. As already pointed out, the effective pressure difference between Ar and O_2_ for the same nominal pressure is ~40% and the van der Waal’s radii are 188 pm for Ar and 152 pm for O [37,38]. A dipole moment should not play a role for a linear molecule such as O_2_, assuming elastic and hard-sphere collisions, which is largely correct for Ar. Therefore, at the same nominal pressure, we expect more collisions in Ar than in O_2_ and the scattering cross section in Ar is larger, thereby underlining the experimentally known observation of a larger stopping power for Ar, as compared with O_2_ [28]. In addition, since there are predominantly elastic collisions in an Ar gas environment, the chemical reactivity in the plasma is considered to be small.

We turn now to the discussion of the results presented in Figure 4. First, in Figure 4a, the comparison of IEDs for Mn^+^ in 2.5 × 10^−2^ mbar Ar and 5 × 10^−2^ mbar O_2_ is shown with both IEDs almost overlapping in intensity and kinetic energy range. For both measurements, the number of background molecules were comparable, i.e., the number of potential scattering events, which indicated that the scattering behaviour of Mn^+^ in both background gases was very similar. Plotting the total integrated intensity of the IEDs for Mn^+^ and Ca^+^ in O_2_ and Ar (Figure 4b,c), which corresponded to the total number of species, the change in area with increasing pressure followed a similar pattern for both ions in both background media, with a similar total number of Mn^+^ and Ca^+^ species for all pressure ranges. After the first initial increase, there was a downturn in the low 10^-2^ mbar range, followed by a steep increase at 5 × 10^−2^ mbar. The main conclusion was that the majority of slowed-down species, irrespective of the background gas, were indeed pushed to small kinetic energies, which strongly suggested that the expansion of Mn^+^ and Ca^+^ in O_2_ and Ar were mainly influenced by non-reactive collisions. On the other hand, in comparing IEDs for Mn^+^ in O_2_ and Ar with a comparable number of background species (see Figure 4a), the total number of species in O_2_ were more than twice as large as compared with Mn^+^ in Ar. This observation indicated that the observed difference was in part due to non-reactive collisions but also included scattering events as a result of chemical reactions during the ablation in O_2_.

This assumption was supported by the pressure dependence of the integrated area of O^+^ ions (Figure 4c). The number of O^+^ ions measured in vacuum corresponded to the oxygen derived from the target, and the same was largely correct for *p* = 1 × 10^−3^ mbar [3]. The strong dip at *p* = 1 × 10^−2^ mbar was the consequence of oxygen being scattered more easily, being the lightest element, whereas the recovery of numbers at larger pressures was the result of the background gas starting to be involved in the chemical reactivity of the plasma. This observation confirmed previous observations where most of the oxygen in *La_0.6_Sr_0.4_MnO_3_* films deposited at a higher deposition pressure originated from the background gas [3]. As for La^+^, the corresponding Ar and O_2_ pressure dependencies showed a decrease in the number of species with increasing pressure, whereas the number of LaO^+^ species increased over the same pressure range by almost two orders. This indicated that the La^+^ species was actively involved in the formation of the LaO^+^ species with increasing O_2_ pressure, and that collisions between La^+^ with *E*_kin_ < 50 eV and the O_2_ background gas molecules led preferentially to the formation of LaO^+^. These measurements further showed that the forward momentum for these heavier species was maintained, which was equally true for La^+^ and LaO^+^. There the energy maximum moved from ~3 eV for a vacuum ablation to >8 eV at 5 × 10^−2^ mbar O_2_ (Figure 4d). At the same time, the energy maximum for LaO^+^ changed from ~3 eV to ~1 eV. The pressure dependence of the LaO^+^ kinetic energy distribution therefore strongly suggested that slow La^+^ species were largely involved in the collision chemistry in the plasma, whereas faster La species seemed to interact via scattering only in a very limited number of cases, and therefore significantly more La^+^ species with a broad energy distribution were detected at an elevated background pressure. The low energy peak for the La^+^ IED in vacuum and the very confined maximum for the LaO^+^ IED at 5 × 10^−2^ mbar (Figure 3d) implied a kinetic energy window of up to 5 eV to be the most effective energy regime to form LaO^+^ species defined by a 90% drop in intensity of the La^+^ signal. Between 5 and 15 eV, the LaO^+^ signal was down by 98% indicating that the majority of LaO^+^ species were accounted for.

Having identified the low kinetic energy range to be favourable for the formation of LaO^+^ at a high O_2_ background pressure, we next studied IEDs for a range of different MO^+^-ions (M = Zr, Y, Ti, Sc, Lu) with dissociation energies larger than 5.12 eV to confirm whether the plasma processes leading to the formation of these species were comparable. Figure 5 shows the IEDs of MO^+^ species selected from Figure 1 measured at different O_2_ pressures (ZrO^+^ = 7.76 eV, YO^+^ = 7.51 eV, TiO^+^ = 6.88 eV, ScO^+^ = 7.11 eV, and LuO = 6.89 eV) [21,39,40]. Irrespective of the element and pressure, all MO^+^ ions showed a maximum in the IED below 10 eV with an increase in the intensity of *E*_kin,max_ with increasing background pressure. This was the same pattern as observed for La^+^/LaO^+^, suggesting a preferential reaction between the background gas molecules and low energy ions. Typical for these MO^+^ binaries with *E*_diss_ > 5.12 eV were the pressure dependences shown in Figure 5a–d. *E*_kin,max_ was observed as shifting to smaller energies with increasing pressure, with a simultaneous increase in the number of species measured. Additionally, for a background pressure of a few 10^−2^ mbar, the IED developed a double peak structure, indicating a slowing-down of as-ablated, and, during the expansion, created MO species. The pressure dependence shown in Figure 5e,f was also frequently encountered, but the origin remains unclear at present. However, the increase in total area with increasing pressure was measurable with ~20% for LuO. As already discussed, the chemical reaction cross section decreased with increasing collision energies [8]. A possible scenario for the gas phase oxidation reaction during the plume expansion in O_2_ for other metal species was therefore analogous to La^+^:(1)Excited, ionised and ground state species collided with O_2_ to generate excited, ionised or ground state species of molecular oxygen (O* or O^+/−^);(2)*E*_kin_ of the ions was reduced due to collisions;(3)Low energy ions reacted with O* or O^+/−^ preferentially to form MO^+^ if *E*_diss_ > 5.12 eV.

To study the gas phase reaction dynamics in more detail, complementary measurements, such as time- and species-resolved imaging, need to be conducted [7,10,28].

## 4. Conclusions

This study gives direct insight into the gas phase oxidation dynamics during the plume expansion of La_0.33_Ca_0.67_MnO_3_ in O_2_ by analysing the kinetic energy distributions of the plasma species at different Ar and O_2_ background pressures. The ion energy analysis by EQP-MS revealed that for M^+^ ions with an MO^+^ dissociation energy smaller than that of O_2_, the expansion in O_2_ was similar to the expansion in Ar, while for ions with an MO^+^ dissociation energy larger than O_2_, the expansion in O_2_ differentiated itself from the expansion in Ar due to the formation of MO^+^. Additionally, there was a favourable energy window where plasma species reacted with the background O_2_, since a significant amount of MO^+^ species with low energies were detected with increasing pressure, as a result of the formation of MO species. The preferred kinetic energy range to form MO species in an expanding plasma plume, as a result of chemical reaction at a high O_2_ background pressure, was approximately up to 5 eV. We have further shown that the chemical activity of species for ablation in an oxygen background was reflected in the kinetic energy profile. Purely inelastic scattering led to a more discreet energy profile with several kinetic energy maxima as a result of excitations, ionisation and dissociation, whereas added chemical activity resulted in a smooth distribution.

## Figures and Tables

**Figure 1 materials-15-04862-f001:**
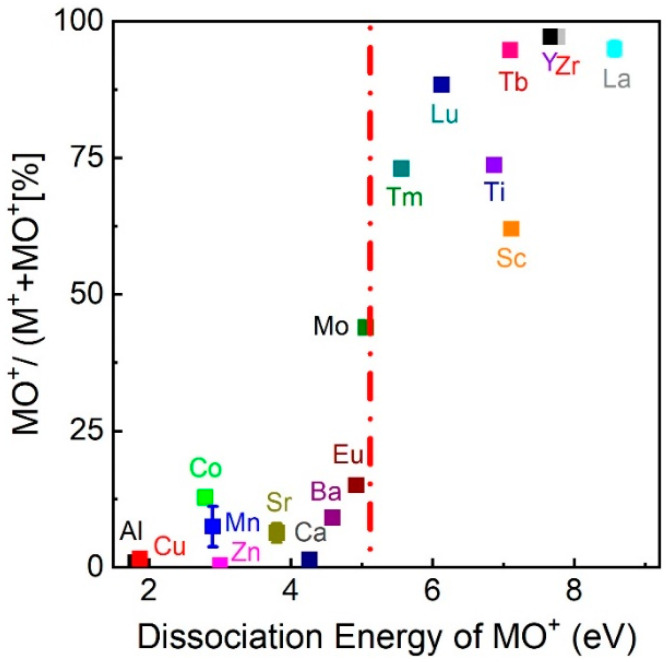
The ratio of MO^+^/(M^+^ + MO^+^) vs. dissociation energy of MO^+^ species as determined at 1.5 × 10^−1^ mbar O_2_. The dashed line represents the dissociation energy of O_2_, $E_{\mathrm{dissoc}}\left(O_{2}\right)=5.12\;\mathrm{eV}$ [9]. Licensee: Christof W Schneider Order Date: 12 July 2022. Order Number: 5346500707826 Publication: Applied Physics Letters.

**Figure 2 materials-15-04862-f002:**
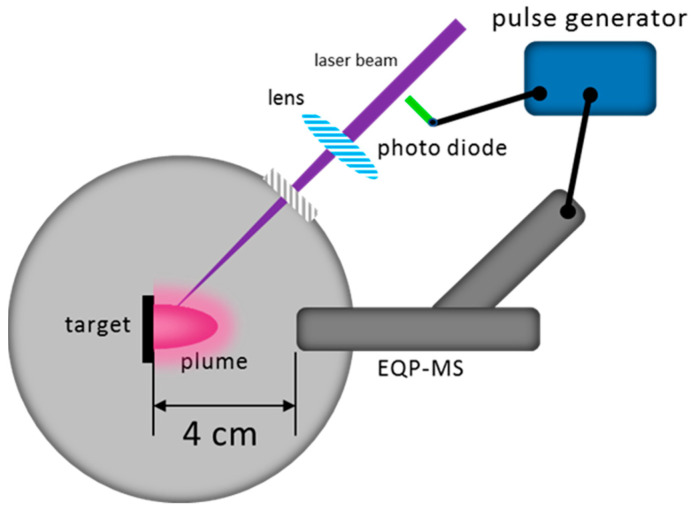
Sketch of the UHV chamber for the ablation and plume characterisation of species at different background pressures and gases.

**Figure 3 materials-15-04862-f003:**
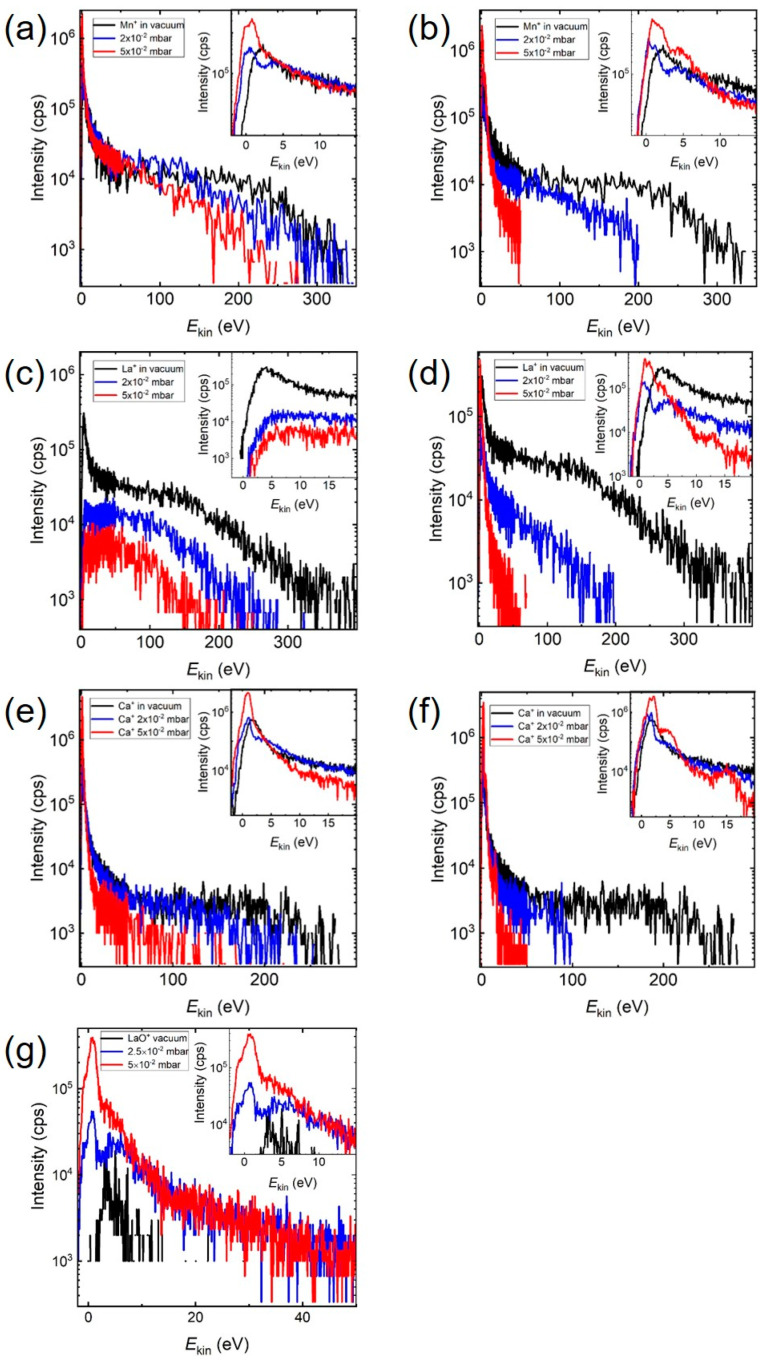
The ion energy distributions for (**a**) Mn^+^ in O_2_, (**b**) Mn^+^ in Ar, (**c**) La^+^ in O_2_, (**d**) La^+^ in Ar, (**e**) Ca^+^ in O_2_, (**f**) Ca^+^ in Ar, and (**g**) LaO^+^ in O_2_, at three selected deposition pressures. The insets show the energy range between 0 and max. 20 eV for the ion energy distribution at different deposition pressures.

**Figure 4 materials-15-04862-f004:**
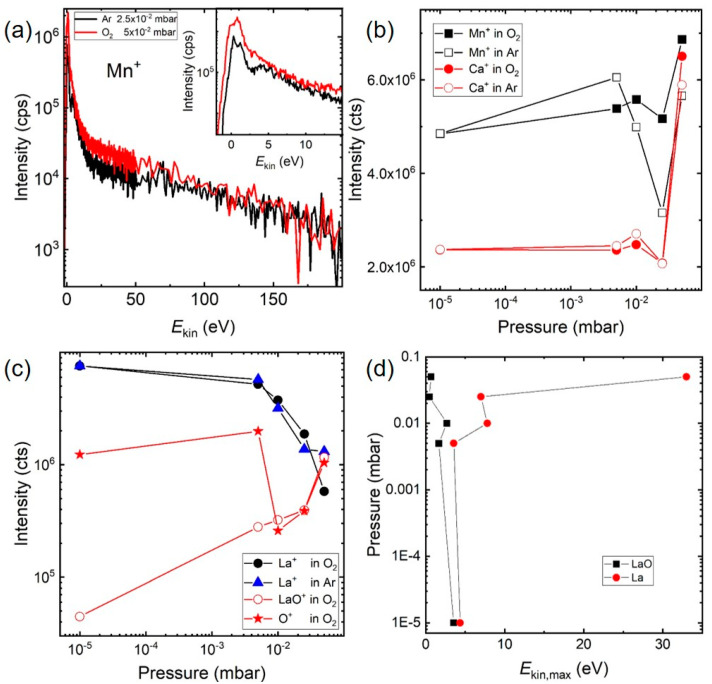
(**a**) Comparison of IEDs for Mn+ in 2 × 10^−2^ mbar Ar and 5 × 10^−2^ mbar O_2_. (**b**) Pressure dependence of Mn^+^ and Ca^+^ in O_2_ and Ar. (**c**) Pressure dependence of La^+^, O^+^ and LaO^+^ in O_2_ and Ar. (**d**) O_2_ pressure vs. *E*_kin,max_ of the ion energy distribution for La^+^ and LaO^+^.

**Figure 5 materials-15-04862-f005:**
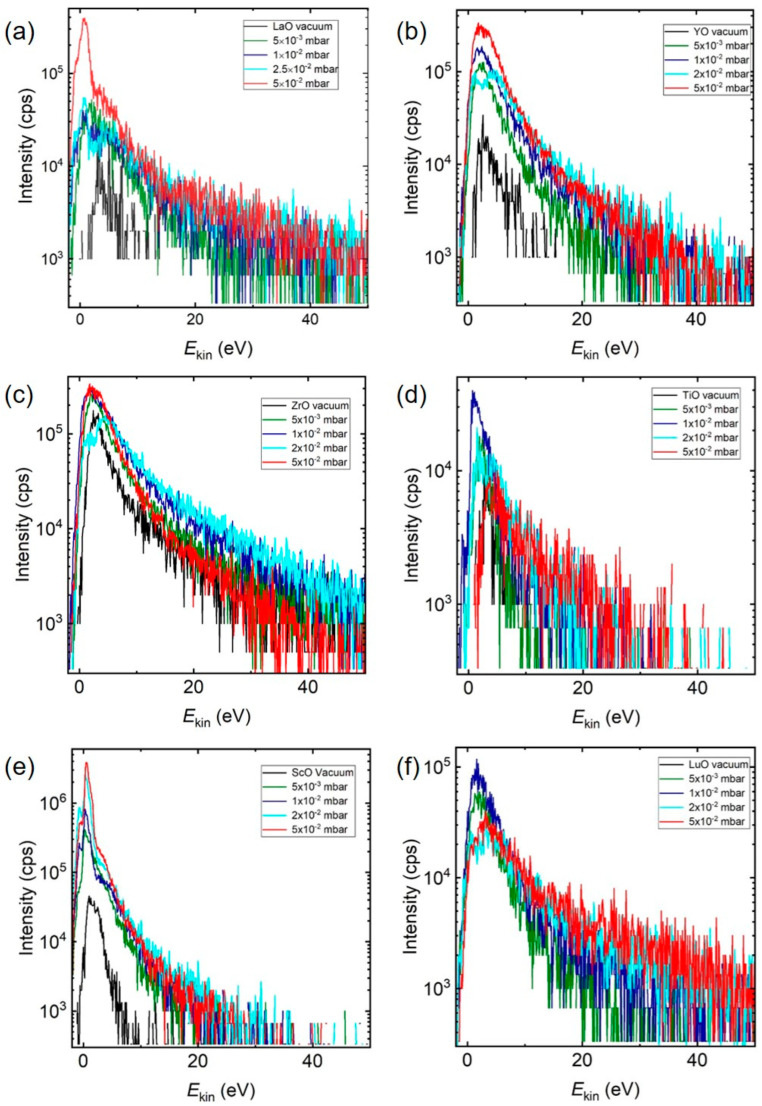
The energy distributions of MO^+^ at different pressures in O_2_ for: (**a**) LaO^+^ from the ablation of La_0.33_Ca_0.67_MnO_3_, (**b**) ZrO^+^, (**c**) YO^+^ from the ablation of 8YSZ, (**d**) ScO^+^ from the ablation of ScMnO_3_, (**e**) TiO^+^ from the ablation of SrTiO_3_, and (**f**) LuO^+^ from the ablation of LuMnO_3_. The ablation conditions are the same for all, with λ = 248 nm and fluence = 2 J/cm^2^.

## Data Availability

Data available on request.

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
