# Peer review of "New Insight into the Gas Phase Reaction Dynamics in Pulsed Laser Deposition of Multi-Elemental Oxides"

_materials, 2022, doi:10.3390/ma15144862_

Round 1
Reviewer 1 Report
Line 10, and throughout text: “Metal” species. Metallic, in my opinion, is a bulk property and does not apply to small species (atoms, etc.).
Lines 22-24: why the “O2 background” gas if the “target composition” is transferred to the film?
Line 43: check sentence starting with equations
Write consistently: Line 38: 1.5 x 10^-1 mbar. Line 45: up to 0.1 mbar. “Up to” or “less than”?
Line 46: mono-atomic instead of “single” ?
Line 48: use equation number instead of “first”.
Line 75: laser repetition rate?
Line 81: which former and latter?
Lines 84-87: what does this mean?
Line 89: provide proper formula of 8YSZ
Figure 2: pulse generator connected to EQP-MS? What for?
Figures 3 and 5: Count rates appear quite high, more than 10^6/s. This should not constitute a problem for DC operation. However, the laser is pulsed and detection takes place during much shorter time intervals. Thus temporary much larger count rates must occur, eventually outside the detection limits. Please comment.
Line 98: chemically inert may be misleading: Ar (excited and metastable) atoms and ions can induce chemical reactions and Ar+ ions may even form molecules, e.g., ArO+, ArM+, Ar2+ dimers, etc, see Rainer Hippler and Christian Denker, 2019 Plasma Sources Sci. Technol. 28 035008, https://doi.org/10.1088/1361-6595/ab0706 and R Hippler et al 2017 J. Phys. D: Appl. Phys. 50 445205, https://doi.org/10.1088/1361-6463/aa8b9a , and references therein, regarding formation mechanisms and pressure dependence.
What about negatively charged ions?
Page 4, first paragraph: I am not sure that a discussion based solely on the dissociation energy is sufficient as, even when energetically allowed, an (additional) dissociation barrier may hinder dissociation.
Line 174: “The reason of these energy windows is not understood …”. There are phrases like: Give me 5 Gaussians and I fit an elephant. In other words: What sense does a fit make if you cannot explain the different components?
Page 6, lines 201/222: It would be nice to learn more about plasma properties, e.g., electron density, electron temperature, atom and ion composition, etc. Do electron reactions (which) take place?
Line 225: phase explosion? instantaneous? Line 226: which substrate in what distance?
Line 226: are neutrals slower? Is the energy distribution known?
Line 233: what does this sentence tell us? How to convert a “number” to a “static (?) pressure” or density?
Line 258-268: these arguments essentially apply to elastic collisions (by the way, mass ratio also plays a role) but not necessarily for inelastic collisions. How large are estimated mean free paths for different species combinations as function of pressure? You could add a table.
Lines 287: more than twice as large, for which pressure?
Line 291: I am not sure that these arguments are valid as reactions taking place on the surface are not considered. How clean is the surface prior and during the laser pulse?
Line 296 in combination with Fig. 4c: I do not buy this argument.
Fig. 4d: what does it show and where is it discussed. Sure that this is something real?
Page 9: I do expect a detailed discussion why you picked MO+ ions and where these ions are generated, i.e., on surface, in hot, or in cold plasma regions?
References. Insert “space” between page numbers and DOI.
Fig. 3 and Supplement: Laser conditions are not provided.
Author Response
We would like to than the referee for his or her comments on our manuscript. A detailed reply is given next.
Line 10, and throughout text: “Metal” species. Metallic, in my opinion, is a bulk property and does not apply to small species (atoms, etc.).
This is correct and has been changed
Lines 22-24: why the “O2 background” gas if the “target composition” is transferred to the film?
The referee is pointing out an interesting misconception with respect to what laser ablation or other physical vapour deposition techniques can do in terms of composition transfer and why a background gas is often used. The main purpose of a background gas for PLD in particular at a higher pressure is to moderate the kinetic energy of the plume species to the extent that most species have a similar average small kinetic energy of a few eV. This is often favourable for the growth kinetics of a film. In the case of O2 it also changes the ratio between metal-oxygen (MO) and single atomic species more in favour of MO species. Again, this is often favourable for the growth of an oxide film. Since oxygen is a light element, it is scattered within a larger opening angle as compared to heavier elements which means that the ratio of light to heavy elements is not easily adjustable for the film growth. This also hold for sputtering. As explained in section 3.3 the physical processes during the initial ablation from the target are responsible for creating a layer of molten material which is not stoichiometric and this includes the oxygen content. Though, there is hardly ever a “correct” composition from the material ablated and the film composition will be adjusted by the ablation conditions like distance, pressure and fluence. The oxygen background typically only supplements but not fully compensates the oxygen “lost” during the ablation and an oxygen annealing step would be required to get to the envisaged oxygen level needed for a correct film composition. There is also one other reason why an oxygen background is used, namely for some materials we have at the deposition pressured used a materials-related oxygen stability line. If you cross it, the material becomes instable and decomposes and one will never be able to prepare them as thin films properly.
Line 43: check sentence starting with equations
The sentence reads correct. There is a comma right after equ. (2), which is part of the sentence.
Write consistently: Line 38: 1.5 x 10^-1 mbar. Line 45: up to 0.1 mbar. “Up to” or “less than”?
This has been corrected
Line 46: mono-atomic instead of “single” ?
This has been corrected
Line 48: use equation number instead of “first”.
This has been corrected
Line 75: laser repetition rate?
The ablation rate was 5 Hz and is now included.
Line 81: which former and latter?
Thanks for pointing the not very precise description. We changed it to O2 and Ar and the sentence reads now as follows: The ion energies were analysed for the ablation of La0.33Ca0.67MnO3 in O2 and Ar with O2 to investigate the gas phase reaction dynamics and Ar as a reference to distinguish the difference between elastic and inelastic collisions of plasma species with the background gas molecules.
Lines 84-87: what does this mean?
It means that we are aware of issues which can affect the dynamics of the plume expansion. As a consequence, but not spelled out, one has to be careful with the interpretation of data. However, as long as the changes in the number of species is not too large, meaning the mean free path is not changed too drastically, we consider the number of scattering events for plasma species in Ar and O2 as similar but certainly not equal and we consider this fact when interpreting data.
Line 89: provide proper formula of 8YSZ
Thanks for pointing out this issue which has been corrected. The sentence reads now as follows: … other oxide targets including 8YSZ (8 mol% Y2O3 fully stabilized ZrO2), BaTiO3, …
Figure 2: pulse generator connected to EQP-MS? What for?
The question of the referee is also touching on another issue with the materials flux and potential saturation of the MS detector. For this reason, we extended the description in the first paragraph of section 2.2. The entire paragraph reads now as follows and the added/changed parts are marked in red:
Ion energy analyses of the plasma plume were performed in a dedicated UHV chamber as shown in Figure 2. An energy-resolved mass spectrometer (Hiden EQP-MS) was installed on the chamber for the ion energy analysis. The Hiden EQP-MS system combines an electrostatic energy analyser and a quadrupole mass spectrometer in tandem to measure the ion energy distributions (IED) for each specific ion. Due to the high materials flux per laser pulse a 30-µm diameter sampling orifice was used and positioned at 4 cm from the target surface normal to the plume expansion axis. In addition, an appropriate tuning of the lenses was done to avoid chromatic aberration and to ensure a stable ion transmission. For the ablation of the targets, a KrF excimer laser (Lambda Physik LPX 300, 20ns, λ = 248 nm, repetition rate: 5 Hz) was used with a fluence of 2 J/cm2 and a laser spot of 1×1 mm2. In order to capture the transient plasma plume, the mass spectrometer was triggered and gated by a pulse generator connected to a photodiode. The pulse generator provides a TTL pulse to the mass spectrometer with a pulse width of 1 ms, which is much longer than the time the plume species require to reach the mass spectrometer. Thus, the mass spectrometer collects all the required species per trigger pulse.
Figures 3 and 5: Count rates appear quite high, more than 10^6/s. This should not constitute a problem for DC operation. However, the laser is pulsed and detection takes place during much shorter time intervals. Thus temporary much larger count rates must occur, eventually outside the detection limits. Please comment.
The referee is pointing out an important issue when working with a pulsed materials flux. For this very reason, but also to be able to run the mass spectrometer at an elevated pressure a 30µm orifice is used.
Line 98: chemically inert may be misleading: Ar (excited and metastable) atoms and ions can induce chemical reactions and Ar+ ions may even form molecules, e.g., ArO+, ArM+, Ar2+ dimers, etc, see Rainer Hippler and Christian Denker, 2019 Plasma Sources Sci. Technol. 28 035008, https://doi.org/10.1088/1361-6595/ab0706 and R Hippler et al 2017 J. Phys. D: Appl. Phys. 50 445205, https://doi.org/10.1088/1361-6463/aa8b9a , and references therein, regarding formation mechanisms and pressure dependence.
Thanks for pointing out this interesting issue and also providing the references. We regularly check the mass spectra for the plasmas created and have not noticed any unusual mass peaks related to the molecules mentioned. We therefore assumed that molecules formed in conjunction with Ar are not relevant if there are any. Looking up bonding energies for these molecules, they seemed to be small and therefore they will not be formed in significant numbers because of dissociation via collisions.
What about negatively charged ions?
For vacuum ablation, there are negative ions but the amount is typically significantly smaller as compared to positive ions and they play some role in the formation of an oxide film. At a higher oxygen pressure, the only relevant neg. ion is O-. Please see also references 9-12.
Page 4, first paragraph: I am not sure that a discussion based solely on the dissociation energy is sufficient as, even when energetically allowed, an (additional) dissociation barrier may hinder dissociation.
The referee raises an interesting and relevant question. From an experimental point of view and as a first order approximations the dividing line is approximately where the dissociation energy of the oxygen molecule is. If this number is exact cannot be determined from the experimental arrangement used. All the metal and metal-oxygen species we have been able to measure fall in these two categories without exception. To discuss a dissociation barrier is certainly interesting and probably relevant if this behaviour can be theoretically described. For the time being this kind of additional discussion is not leading to a new insight without experimental evidence to prove this point.
Line 174: “The reason of these energy windows is not understood …”. There are phrases like: Give me 5 Gaussians and I fit an elephant. In other words: What sense does a fit make if you cannot explain the different components?
The referee is highlighting a specific issue related to the limits of the experimental technique used. Mass spectroscopy will not tell you anything about specific reaction kinetics taking place in the plasma plume, which species are created temporarily and which species are involved. The measurement is mapping the integral of the kinetic energy of the arriving species many µsec after the initial ablation process has been started. Details of events in between can only be indirectly deduced, not measured nor pinpointed. Identifying a range of kinetic energies which can be potentially associated with a certain reaction kinetic and dynamic is already an achievement and not reported before to the best of our knowledge.
Page 6, lines 201/222: It would be nice to learn more about plasma properties, e.g., electron density, electron temperature, atom and ion composition, etc. Do electron reactions (which) take place?
With the techniques used for this study, electron density or temperature cannot be measured. This would require a Langmuir-setup to measure the electron and/or ion current inside the plasma. This has been done for other systems and data are published by us and many other groups (https://doi.org/10.1063/1.3056131, https://doi.org/10.3390/coatings11070762, https://doi.org/10.1016/j.apsusc.2017.03.055, https://doi.org/10.1016/j.apsusc.2012.12.090, https://doi.org/10.1063/1.4815989, https://doi.org/10.1063/1.3516491). Usually, the analysis using ion probes works very well for metallic targets, but reliable date for multi-elemental targets is extreme difficult or even not possible. In principle, these properties could also be deduced from optical emission spectroscopy (https://aip.scitation.org/doi/10.1063/1.2338282, https://link.springer.com/article/10.1007/s11666-011-9681-6, https://doi.org/10.1016/j.apsusc.2007.02.032). Again, multi-elemental systems are very difficult to treat. Likewise, the composition of a plasma is known for some systems and data are published. To characterize all systems the way the referee suggests is beyond the intention of the manuscript and more useful for a review rather than for a specific topic.
Line 225: phase explosion? instantaneous? Line 226: which substrate in what distance?
We apologize for not mentioning in the preceding paragraph the time scale of the initial evaporation when the laser light starts to interact with the target material. It is of the order 10 psec, which is short compared to the length of the laser pulse (20 nsec) and the time required for the plume to travel 4 cm, a typical target-substrate distance. The terminology for this violent very short-term evaporation is “phase explosion”. We also should not have used the word substrate. What we meant was the arrival of species at the MS location which is equivalent to a heater position where one could mount a substrate. The sentence was changed and reads now: For a vacuum ablation, fast species reach the position of the mass spectrometer or heater, respectively, within 1 µsec as measured using time resolved emission spectroscopy.
Line 226: are neutrals slower? Is the energy distribution known?
The referee assumes correctly that neutral species are slower than ions and the energy distribution seems similar to the ones known for ions. However, measuring the energy distribution is much more difficult by mass spectrometry as compared to ions since the ionization cross section of the neutral species is influenced by their kinetic energy. In addition, the geometrical boundary conditions for a mass spectrometer, in particular with the very small aperture used, makes it even more demanding to measure neutrals with a large kinetic energy correctly. Measuring neutral requires at the aperture of the MS to apply a sufficiently large electric field to repel the ions and let neutral only pass. These fields will influence the plume with unknown consequences for the kinetic energy distribution. In other words, we do not really know what quantity is going to be measured when trying to measure the kinetic energy distribution of neutrals.
Line 233: what does this sentence tell us? How to convert a “number” to a “static (?) pressure” or density?
The sentence in line 233 has been change as already mention previously.
We did not convert the number of species removed per pulse into a static pressure. It is only stated that the number of species removed per pulse would be equivalent to a pressure of … . The amount of material removed per pulse (20nsec) is roughly 1mm x 1mm x several 10nm depending on the absorption properties of the surface layer. This number can be scaled into a volume occupied after a certain amount of time after the initial ablation. From time-resolved measurements one can get a good idea about the initial opening angle and how quick species move and hence how the nominal density would change with time and distance.
Line 258-268: these arguments essentially apply to elastic collisions (by the way, mass ratio also plays a role) but not necessarily for inelastic collisions. How large are estimated mean free paths for different species combinations as function of pressure? You could add a table.
The nominal mean free path for the different species is somewhere between several 10th of cm at a low pressure and mm and shorter for the 10-1 mbar range. That these numbers cannot be really correct is evident because of the time window a plume can be observed in vacuum (2-3µsec after the laser impacts on the target at a distance 4cm away from the target). The observed optical excitations have a nominal lifetime, depending on the species, of a couple of ns up to several hundred nsec. This only works if plume species are re-excited due to scattering but this requires a certain species density and velocity. A mfp with cm distances doesn’t really work well. To emphasis this point we added the following to the text:
A rough estimate of the number of species removed from the target with one pulse corresponds approximately to the number of atoms equivalent of a static pressure in the mbar range (1021 - 1022/m3). As a consequence of such an initial high local density the mean free path (mfp) becomes very short (« mm) and even at the time of arrival at the location of the mass spectrometer the plume density must be large enough to enable a sufficient number of re-excitations via scattering of optically excited species since the typical lifetime of excited states ranges from nsec to several hundred of nsec. This plume related mfp is still much shorter than a nominal mfp of meters for a pressure of 10-5 mbar and below and the number of scattering events until species reach the mass spectrometer is at least 10 to 20.
Lines 287: more than twice as large, for which pressure?
Thanks for pointing out this issue. The Ar pressure should read 2.5 x10-2 mbar, O2 5x10‑2 mbar. Here, we compared two measurements where we have been trying experimentally to take care of this issue with the nominal and real pressure. We should have made this more clear in the text. The sentences in question starting in line 280 reads now as follows: “First, in Figure 4 a) the comparison of IEDs for Mn+ in 2.5x10-2 mbar Ar and 5x10-2 mbar O2 is shown with both IEDs almost overlap in intensity and kinetic energy range. For both measurements the number of background molecules is comparable i.e., the number of potential scattering events, which indicates that the scattering behaviour of Mn+ in both background gases is very similar.”
Line 291: I am not sure that these arguments are valid as reactions taking place on the surface are not considered. How clean is the surface prior and during the laser pulse?
The comment by the referee is probably a consequence of the substrate mentioned earlier. Mass spectroscopy does not provide any information on chemistry related to material deposited on a substrate surface. Hence, we cannot comment on this part. Also, the discussion on the kinetic energy distributions is independent of what happens on a substrate.
Line 296 in combination with Fig. 4c: I do not buy this argument.
It is not fully clear with which parts of the arguments the referee disagrees with or is not convinced. If we would plot in Fig. 4c Ekin,max instead of the integrated intensity, the plot would look not much different and the conclusions remain the same. Also, for the pressure dependence of O+, the experiments presented in ref. 3 where we worked with an 18O-exchanged La0.6Sr0.4MnO3 target, show where the oxygen in a film, must come from.
Fig. 4d: what does it show and where is it discussed. Sure that this is something real?
The answer to the referee’s question is given starting in line 313: “These measurements further shows that the forward momentum for these heavier species is maintained which is equally true for La+ and LaO+. There the energy maximum moved from » 3 eV for a vacuum ablation to > 8 eV at 5 × 10-2 mbar O2 (Figure 4d)….” To place a very definitive value for a maximum to measurements as shown in Fig. 3b insert, is challenging. Still the general tendency is accurate.
Page 9: I do expect a detailed discussion why you picked MO+ ions and where these ions are generated, i.e., on surface, in hot, or in cold plasma regions?
The reason why we study MO+ species is stated in line 328 and following. We want to know if the metal species with a large binding energy to form MO+ is similar or dissimilar as a function of oxygen pressure. The referee is expecting a discussion on the generation of MO+ ions, in particular where they are created in the plasma. This is a relevant question, but in the context of the used measurement technique (energy resolved mass spectroscopy) a question the technique cannot provide an answer a priori. To get closer to this question requires a time and spatially resolved technique which traces specific ionic optical excitations. There are only a very limited number of measurable MO II (with II meaning single ionized) excitations which can be found and species like LaO II are difficult to measure time and spatial dependent because these signals are very weak.
References. Insert “space” between page numbers and DOI.
Thanks for pointing out this issue. The style issue with Endnote has been resolved.
Fig. 3 and Supplement: Laser conditions are not provided.
The ablation conditions used for all experiments are detailed in section 2.

Reviewer 2 Report
The work is generally interesting for those use PLD techniques and gives insight into how the species react in the plume under different pressure. However, the manuscript should be improved significantly.
-The discussions should be summarized as many statements were repeated more than once throughout the texts. The main points should be explained only and justified briefly.
-The discussions should be linked to some PLD equations and matter-light interaction theory related to the PLD process.
-I recommend that the authors provide the plume shape for the distribution shown in Figure 3, which will offer more value for the work.
-Finally, this study showed interesting results, but if the authors can show XRD or SEM measurements of the optimized conditions, This will provide how the distribution, plume shape, and film morphology can be linked.
Author Response
We would like to than the referee for his or her comments on our manuscript. A detailed reply is given next.
The work is generally interesting for those use PLD techniques and gives insight into how the species react in the plume under different pressure. However, the manuscript should be improved significantly.
-The discussions should be summarized as many statements were repeated more than once throughout the texts. The main points should be explained only and justified briefly.
Thanks for the suggestion and we change the text in the discussion section 3 accordingly.
-The discussions should be linked to some PLD equations and matter-light interaction theory related to the PLD process.
We thank the referee for the suggestion to extend the discussion to include matter-light interaction theory and some basic PLD equations. In the general context of the manuscript, the observation of the kinetic energy distributions of ions as detected by ion mass spectroscopy, we do not see the benefit of this kind of discussion which would be much more related to the initial ablation process. The information about initial processes is not provided by mass spectroscopy. It is about what happens in a plasma in particular in connection with a background gas at different deposition pressure, it is about the deduction of “rules” governing the plume species interaction before they condense on a substrate to form a film. This is of particular interest for the deposition of multi-elemental materials and plume species with very different masses.
-I recommend that the authors provide the plume shape for the distribution shown in Figure 3, which will offer more value for the work.
Thanks for the suggestion to include the plume shape of the different materials shown in Fig. 3 for the general discussion. We regret to say, this cannot be done for the following reasons. The plume shape is typically looked at using (time-resolved) optical spectroscopy. For this technique, optically excited species are measured. Most excited species are neutral species, excited ionic species are scarce and their intensity is often very weak. Hence ionic species are difficult to trace if one attempts to measure them time, spatially and frequency resolved. The manuscript discusses ions, imaging is most of the time about neutrals and both species do have somewhat different expansion properties in a plume. One can image the plume just time and species integrated, but than properties of neutrals and ions are mixed. One can image plume species (and time) resolved, but there the expansion properties depend on the masses of the imaged species. The question is, what is the value of including a plume-shape in a discussion whereas mass spectroscopy is providing data of ions just from the forward direction? A discussion with plume shape is meaningful if we would discuss an angular dependence also measured with ion mass spectroscopy, something which is not a topic of this manuscript.
-Finally, this study showed interesting results, but if the authors can show XRD or SEM measurements of the optimized conditions, This will provide how the distribution, plume shape, and film morphology can be linked.
Thanks for the very useful suggestion to include data on thin films to the manuscript. This has not been done for the following reasons. First, the manuscript is about the analysis of plasma conditions, not about thin film growth. We also did not state explicitly, that these are optimized conditions for a particular material deposited as thin film. The used deposition conditions are very typical for oxide thin film growth and hence rather general. Second, if we would include film growth, we should do it for all the materials we looked at. This would unquestionably be a very extensive manuscript since we would have to study the composition as well, and if feasible deposition temperature dependent. We looked into the room temperature, angular dependence of the composition in the past (https://doi.org/10.1002/admi.201701062) and these films have been amorphous. The link between film morphology and mass spectroscopy is therefore already established and documented. The more relevant high deposition temperature studies cannot be done. However, the plasma chemistry will not change irrespective of the deposition temperature.

Reviewer 3 Report
The paper is focused on the investigation of the gas-phase reaction dynamics and kinetics in a laser induced plasma, when metal and metal-oxygen species were ablated. It has been revealed that the content and pressure of the “background” gas influence strongly these processes. Also, the authors suggest that the dissociation energy of the charged metal-oxygen species determines the character of the expansion.
The article may be of interest for the researches working on deposition of metal and metal oxide films.
I would recommend acceptance of the paper after the minor revision.
Please find below some questions and remarks:
1. P.5 fig 3, lines 162 and so on.
Couldn’t you comment in details why the ion energy distribution has 3 maxima (as indicated, at <5eV, at 5-10 and at 10-20 eV)? How do you attribute these 3 peaks? It seems that there is also the 4th maximum at around 150 eV (for example, Fig.3b and f).
2. Is it possible to change the distribution of ion energy (to shift the maxima and/or vary the width) by varying the parameters of the laser irradiation (laser fluence, wavelength, pulse duration etc.)? Does the probability of the chemical reactions depend on the initial ablation and plasma formation?
3.Some of the references were not reproduced in the pdf file. Here is the list of lines where it took place:
p.2 line 69
p.3 lines 95 and 110
p.4 line 147
p.5 line 186
p.6 line 244
p.8 lines 273 and 310
p.9 lines 324 and 325
4. Also please find below some misprints and grammar mistakes:
p.5 line 162 positiv->positive
p.8, lines 305-306 ”that collisions between … leads”-> ”that collisions between … lead”
p.8, line 307 “These measurements further shows”-> “These measurements further show”
Author Response
We would like to than the referee for his or her comments on our manuscript. A detailed reply is given next.
The paper is focused on the investigation of the gas-phase reaction dynamics and kinetics in a laser induced plasma, when metal and metal-oxygen species were ablated. It has been revealed that the content and pressure of the “background” gas influence strongly these processes. Also, the authors suggest that the dissociation energy of the charged metal-oxygen species determines the character of the expansion.
The article may be of interest for the researches working on deposition of metal and metal oxide films.
I would recommend acceptance of the paper after the minor revision.
Please find below some questions and remarks:
- P.5 fig 3, lines 162 and so on.
Couldn’t you comment in details why the ion energy distribution has 3 maxima (as indicated, at <5eV, at 5-10 and at 10-20 eV)? How do you attribute these 3 peaks? It seems that there is also the 4th maximum at around 150 eV (for example, Fig.3b and f).
Thanks for the comment. Currently, we don’t have evidence or an indication about the potential origin of these peaked structures. Typically, a two or three, sometimes a four-peak Boltzmann-distribution fit is sufficient to fully describe the energy dependencies measured. A first order assumption could be that the kinetic energies are cascaded down from high to low energies due to scattering events. These scattering events will depend on the initial kinetic energies of these species, slower species scatter more readily than faster ones. Species are also created at different times during the initial ablation window which is 20nsec plus the time needed to have the molten material of the target solidified giving rise to very different kinetic energy profiles even for the same type of plasma species. Modelling is required to gain a better insight which of the scattering processes are favourable.
- Is it possible to change the distribution of ion energy (to shift the maxima and/or vary the width) by varying the parameters of the laser irradiation (laser fluence, wavelength, pulse duration etc.)? Does the probability of the chemical reactions depend on the initial ablation and plasma formation?
The referee raises several valid points. At present we do not know the answer to the questions put forward. Ablation parameters like fluence and wavelength will change the distributions. With both parameters one is effectively changing the ablation rate due to differences in the absorption and photochemistry hence the number of species will change for the ablation. This will in turn change the scattering behaviour in the plume. Using a different wavelength changes also the photodissociation of species and hence the composition of the plume. We would therefore expect to see a different kinetic energy profile for 193, 248 and 308nm for the same material. Changing the pulse length cannot be done easily for excimer lasers which is given by the thyratron and the gas mixture.
3.Some of the references were not reproduced in the pdf file. Here is the list of lines where it took place:
p.2 line 69, p.3 lines 95 and 110, p.4 line 147, p.5 line 186, p.6 line 244, p.8 lines 273 and 310, p.9 lines 324 and 325,
We thank the referee pointing out this issue, which has been resolved.
- Also please find below some misprints and grammar mistakes:
p.5 line 162 positiv->positive
p.8, lines 305-306 ”that collisions between … leads”-> ”that collisions between … lead”
p.8, line 307 “These measurements further shows”-> “These measurements further show”
We thank the referee pointing out this issue, which have been resolved.

Reviewer 4 Report
At this stage of my review, I wish to point out the following important points (both of physics and organisation of the manuscript) that must be addressed before the manuscript could be considered for publication.
- The title is very general and suggests the paper is a major review of this topic (of long interest in the field of laser ablation/PLD). I would recommend that the authors reconsider the title to make it more specific and better reflect the work and data presented in the paper.
- The intro clearly presents the problem and notably the competing M+ vs MO+ reaction paths, substantiated by a large amount of data (Fig.1). However, the key sentence (lines 54 and 55) that exposes the problem being tackled in the work is about formation of "MO species", ie the neutral molecule. How is this related to the M+ vs MO+ problem? What is the supplementary reaction path after (1) that leads to the neutral? Molecular ions can be extremely stable molecules and the reasons why and how they would neutralise are not explained. This part of the paper needs to be revised and the physics/chemistry connection to the neutral molecule must be made explicitly.
- What type of energy is meant by "energy window"? Kinetic energy (KE)? This is unclear since the KE is always a distribution. Ultimately the peak of the KE distribution is controlled by the laser fluence at a given pressure and this is the control parameter. This needs to be explained by the authors.
- The key physical concepts, highly relevant to the present work (see abundant previous literature), of hydrodynamic vs molecular flow, Knudsen flow, mean free path (compared to size of vacuum vessel) are not mentioned at any stage in the manuscript. This needs to be addressed
- Figure 3 is unreadable -far too many small panels. It should be redone.
- Figs 3, 4(a) and S1 (and Fig.5 to some extent) are very badly presented. They essentially show only the noisy data at large KE's, that's of very limited use, whereas the meaningful data (at the lower KE's) is shown in the inserts only. This is due to the choice of scales in the figures by the authors. The authors must redo these figures to present their data correctly and meaningfully.
Author Response
We would like to than the referee for his or her comments on our manuscript. A detailed reply is given next.
At this stage of my review, I wish to point out the following important points (both of physics and organisation of the manuscript) that must be addressed before the manuscript could be considered for publication.
The title is very general and suggests the paper is a major review of this topic (of long interest in the field of laser ablation/PLD). I would recommend that the authors reconsider the title to make it more specific and better reflect the work and data presented in the paper.
We thank the referee for the suggestion to modify the title of the manuscript. It reads now: “New insight into gas phase reaction dynamics in pulsed laser deposition”
The intro clearly presents the problem and notably the competing M+ vs MO+ reaction paths, substantiated by a large amount of data (Fig.1). However, the key sentence (lines 54 and 55) that exposes the problem being tackled in the work is about formation of "MO species", ie the neutral molecule. How is this related to the M+ vs MO+ problem? What is the supplementary reaction path after (1) that leads to the neutral? Molecular ions can be extremely stable molecules and the reasons why and how they would neutralise are not explained. This part of the paper needs to be revised and the physics/chemistry connection to the neutral molecule must be made explicitly.
We thank the referee for the issue raised. What we meant in line 54/55 are the formation of MO species in the plasma in general, neutral and ionic. To make this explicit the sentence reads now: …the question of interest is to determine the favourable kinetic energy window at which plasma species react/interact with the background gas to form neutral and ionic MO species.
The mechanism leading to neutrals in the plasma is probably electron detachment via scattering in most cases. It would also be possible to obtain an electron from the electron cloud in front of the plume when the ionic species at some point catch up. In case the referee is wondering why we did not present MO data since they can also be measured with the MS. The reason is, in order to measure neutrals an electric field at the aperture has to be set to prevent the ions entering the orifice. The large electric field required will affect the arriving plasma species and one has no means to gauge the outcome. In addition, the ionization is KE dependent. We tried to measure KE distributions of neutral in the past but the results have not been convincing.
What type of energy is meant by "energy window"? Kinetic energy (KE)? This is unclear since the KE is always a distribution. Ultimately the peak of the KE distribution is controlled by the laser fluence at a given pressure and this is the control parameter. This needs to be explained by the authors.
We thank the referee for pointing out the inaccuracy. Since the context is kinetic energy we thought it to be obvious that the use of energy and kinetic energy is equivalent.
Is the referee certain that a kinetic energy is a distribution? A kinetic energy per se is a number.
We agree with the referee that the peak positions in the distributions will be influenced and controlled by the fluence and likewise the laser beam cross section (and the wavelength). The number of species involved in scattering processes will be the determining parameter.
The key physical concepts, highly relevant to the present work (see abundant previous literature), of hydrodynamic vs molecular flow, Knudsen flow, mean free path (compared to size of vacuum vessel) are not mentioned at any stage in the manuscript. This needs to be addressed
The referee raises an interesting point with respect to the mean free path. We included this now in the manuscript and the added sections reads as follows:
A rough estimate of the number of species removed from the target with one pulse corresponds approximately to the number of atoms equivalent of a static pressure in the mbar range (1021 - 1022 / m3). As a consequence of such an initial high local density the mean free path (mfp) becomes very short (« mm) and even at the time of arrival at the location of the mass spectrometer the plume density must be large enough to enable a sufficient number of re-excitations via scattering of optically excited species since the typical lifetime of excited states ranges from nsec to several hundred of nsec. This plume related mfp is still much shorter than a nominal mfp of meters for a pressure of 10‑5 mbar and below and the number of scattering events until species reach the mass spectrometer is at least 10 to 20. This would also imply that even at a distance 4 cm away from the target, the hydrodynamic conditions are more viscous rather than molecular.
Figure 3 is unreadable -far too many small panels. It should be redone.
Figs 3, 4(a) and S1 (and Fig.5 to some extent) are very badly presented. They essentially show only the noisy data at large KE's, that's of very limited use, whereas the meaningful data (at the lower KE's) is shown in the inserts only. This is due to the choice of scales in the figures by the authors. The authors must redo these figures to present their data correctly and meaningfully.
We thank the referee for his or her comments about the figures. That some of the data are more noisy is to some extend a misconception. Some of the signals measured are small, indeed, and we show the entire data set as measured. That the energy-resolution varies is in parts an instrumental issue. At higher voltages the resolution of the MS (voltage step-width) it is smaller. The data presented are as measured even if the amount of information is in places somewhat limited. We agree on this point. We have been trying to keep the energy scales, also for the insert, in Figures 3, 4, 5 and S1 as similar as possible. This should help a reader to compare data more easily, often an issue encountered in publications that this cannot be done.
After consulting with the editor, we decided to use a two-column format to present the data. If presented on A4, the panels are well readable.

Round 2
Reviewer 1 Report
The paper is improved and now ready for publication.
Author Response
We would like to thank the referee for recommending the manuscript for publication.

Reviewer 2 Report
The manuscript improved significantly compared to the original one, and the reference citation has been used sufficiently to justify the authors' claims. However, the authors did not addressed all comments. I still recommend if possible showing the photo of the ion plume for different ions using the same energy as well as the photos of the plume for one ion but with different energies. The plume shape is also important when we study such distribution in PLD technique. I recommend citing the following PLD paper that shows how the energy of the landed ions is affected by the pressure and how this influence significantly grown material structures: RSC advances 5 (115), 94670 (2015).
Author Response
We would like to thank the referee for his or her comments. A detailed reply is given separate.
Reviewer 4 Report
(1) The authors have partly addressed the issue of the title of the paper, but it still is not precise enough. To make my point a little bit better, I will suggest this title: "New insights into the gas phase reaction dynamics in pulsed laser deposition of metal oxides". The authors will decide ultimately.
(2) The question about MO vs MO+ is not really addressed from the physics viewpoint but the changes in the manuscript read OK. (By the way, in their reply letter, the authors mention "electron detachment" as the likely mechanism for neutralisation of the MO+ molecules? Since this is not mentioned in the actual text of the manuscript, we won't discuss this point any further).
(3) Question about the "Kinetic energy": The authors' reply sounds a tad unwarranted. Indeed, yes, the referee is absolutely certain that the kinetic energy (not a "number per se", but a physical quantity with the units of J) of the laser ablated ions follows a distribution (that of v^2 obviously). The shape and peak maximum of the distribution will be determined by the laser energy for a given spot size (typically fixed in a given deposition set up) and laser wavelength (always UV in pld). Therefore, the laser energy is the control parameter in a given experimental configuration and this was the basis of my question, ie that the "favourable energy window" needed to be qualified.
(4) Presentation of data/figures. The use of a full page for Figure 3 is welcome. However, the more fundamental point of the choice of the horizontal energy (Ekin) scale has not been properly addressed. The authors' meaningful data is the one showed in the inserts, ie in the 0- 20 eV (low) KE range. This is the (significantly above the noise) data that will mostly determine the value of the parameters of the shifted Maxwellian distributions (see Section 3.2). The high energy data (50 - 350 eV) is the high energy tail of the distributions, with orders of magnitude (3 or 4) lower counts and therefore significantly decreased signal-to-noise ratios (visually enhanced by use of log scale). The inserts should become the main parts and vice-versa (full energy range data can be shown in an insert). Figure 5 is more meaningful in that respect and the reader could estimate KE peaks and widths with a much better precision from the figure.
(5) In addition, in all the inserts of Figure 3, the traces extend significantly to the left of the tick of zero KE (ie as if there were negative KE's?). This must be addressed.
(6) The authors must show at least one example, taken from Figure 3, of how the observed KE distributions can be fitted by " two or three MB distributions", ie superimpose the best fits onto the data.
Author Response

(The authors gave the same response as above.)
